# Therapeutic Potential of Annexin A1 Modulation in Kidney and Cardiovascular Disorders

**DOI:** 10.3390/cells10123420

**Published:** 2021-12-05

**Authors:** Mahmood S. Mozaffari

**Affiliations:** Department of Oral Biology and Diagnostic Sciences, The Dental College of Georgia, Augusta University, Augusta, GA 30912-1128, USA; Mmozaffa@augusta.edu

**Keywords:** annexins, kidney, cardiovascular system, inflammation, acute kidney injury, nephropathy, glomerulonephritis, renal cancer, kidney stone, atherosclerosis, myocardial infarction

## Abstract

Renal and cardiovascular disorders are very prevalent and associated with significant morbidity and mortality. Among diverse pathogenic mechanisms, the dysregulation of immune and inflammatory responses plays an essential role in such disorders. Consequently, the discovery of Annexin A1, as a glucocorticoid-inducible anti-inflammatory protein, has fueled investigation of its role in renal and cardiovascular pathologies. Indeed, with respect to the kidney, its role has been examined in diverse renal pathologies, including acute kidney injury, diabetic nephropathy, immune-mediated nephropathy, drug-induced kidney injury, kidney stone formation, and renal cancer. Regarding the cardiovascular system, major areas of investigation include the role of Annexin A1 in vascular abnormalities, atherosclerosis, and myocardial infarction. Thus, this review briefly describes major structural and functional features of Annexin A1 followed by a review of its role in pathologies of the kidney and the cardiovascular system, as well as the therapeutic potential of its modulation for such disorders.

## 1. Introduction

Annexins were discovered over 40 years ago and constitute a superfamily of more than a thousand proteins, in major eukaryotic phyla, with complex and diverse functions including calcium metabolism, intracellular signaling, subcellular transport, membrane repair, cell adhesion, cell death/survival, growth and differentiation, among others. Structurally, the C-terminus of annexins is comprised of four homologous repeats of about 70 amino acids which contain the endonexin sequence (GxGT-38 residues) that serves as a (type 2) calcium binding site with high affinity for calcium and phospholipids. The variable N-terminus harbors sites for post-translational modifications and protein-protein interactions [1,2,3]. Although substantial homologies exist among the superfamily of annexins, variations in amino acid sequences among different groups of eukaryotes translate into structural characteristics and biochemical features causing eukaryotic group-specific functional differences. The current nomenclature for annexins takes into consideration the evolutionary divisions of the eukaryotes and refers to those of the vertebrates, including human, as family A with the 13 annexins commonly found in vertebrates named as Annexin A1 to A13 (also known as ANXA1-ANXA13) [1,2,3,4,5,6]. Human annexin-A7 (synexin) was the first to be isolated and purified and human Annexin A1 (lipocortin) and -A2 (calpactin) were the first to be cloned [1,2].

Annexin A1, the product of *ANXA1* gene, is a 37 kDa protein that was initially discovered (in the 1980s) as a dexamethasone (DEXA)-inducible protein (e.g., in renal cells). It has also been referred to as lipocortin-1, renocortin, and macrocortin [5,6]. It is now known that Annexin A1 is expressed in variety of tissues (e.g., heart, brain, blood vessels) and that immune cells (e.g., leukocytes) display high expression levels. At the cellular level, Annexin A1 is expressed in the nucleus, cytoplasm, and cell membrane, consistent with its multifaceted functions (e.g., proliferation, apoptosis, survival, differentiation and migration, anti-inflammatory, etc.) [7]. The ability of glucocorticoids to regulate mRNA transcription of Annexin A1 has been shown in in vitro (e.g., using a human epithelial cell line A549), ex vivo (e.g., alveolar macrophages of patients with inflammatory lung disease), and in vivo (e.g., rat brain) studies [8]. Importantly, aside from transcription, DEXA also causes the translocation of Annexin A1 to the cell membrane and its secretion, processes likely mediated through serine-27 phosphorylation via mitogen-activated protein kinase- (MAPK), phosphatidylinositol-3-kinase- (PI3K), and protein kinase C-dependent signaling pathways (e.g., in a human folliculostellate cell line) [9]. Importantly, cell-specific molecular events culminating in the secretion of Annexin A1 are of major relevance and implication for the binding and attachment of leukocytes onto endothelial surfaces [8]. For example, ATP binding cassette transporter system is believed to regulate Annexin A1 secretion in macrophages while in neutrophils cytoplasmic Annexin A1 is stored in gelatinase granules and is released in response to chemoattractants or adhesion to endothelial cells monolayers [10]. Aside from its potent inhibitory effects on neutrophil trafficking, several other major effects of Annexin A1 on leukocytes have been described, including the (a) release of Annexin A1 from apoptotic neutrophils promoting macrophage phagocytosis and removal of apoptotic cells, (b) Annexin A1 secretion by macrophages (e.g., in response to glucocorticoids), acting in autocrine or paracrine fashion, increasing entrapment of apoptotic neutrophils, and (c) exogenous administration of Annexin A1 increasing human neutrophil apoptosis mediated, in part, via dephosphorylation of pro-apoptotic protein- Bad [10].

One of the early discoveries which subsequently led to further unraveling of the anti-inflammatory effects of Annexin A1 is its inhibition of cytosolic phospholipase A2 (PLA2) and subsequent blockade of arachidonic acid release leading to the inhibition of generation of eicosanoids (e.g., prostaglandins such as PGE2) [5,11,12,13,14,15,16]. Further, as indicated above, Annexin A1 is also transported to the cell membrane and externalized or secreted into the extracellular fluid which, in turn, exerts its effects (e.g., anti-inflammatory) [7]. Importantly, the myriad of effects attributed to Annexin A1, as well as its N-terminal-derived peptides (i.e., Ac2-26, Ac2-12, and Ac2-6), are believed to be mediated via binding to G-protein coupled receptors, e.g., formyl peptide receptors 1 and 2 (Fpr1, Fpr2), and/or its ability to bind to cell membrane phospholipids [8,17,18]. Fpr2 is the main receptor for anti-inflammatory effects of Annexin A1. It is also known as the ALX as it also serves as the receptor for the anti-inflammatory molecule lipoxin A4 (i.e., Fpr2/ALX) [10]. Accordingly, extracellular Annexin A1, in the presence of ≥1 mM Ca^2+^, undergoes conformational change that leads to exposure of the N-terminal region and binding thereby facilitating its interaction with Fpr2/ALX. Annexin A1 can function in an autocrine, paracrine, and juxtacrine (i.e., involving cell-cell contact) fashion to activate Fpr2/ALX signaling; the latter mechanism is suggested as the most plausible mechanism for the actions of Annexin A1 in inflammatory conditions [10]. The Annexin A1-induced Fpr2/ALX signaling involves phosphorylation of extracellular-regulated kinase (ERK) 1/2 that is associated with rapid elevation in intracellular calcium concentration. Indeed, blockade of Fpr2/ALX signaling, using a monoclonal antibody, inhibited Annexin A1-induced human neutrophil transmigration and adhesion to endothelial-cell monolayers under flow [19].

The early recognition of the role of Annexin A1 as an anti-inflammatory protein has been followed by the unraveling of a myriad of its effects, with therapeutic potential, in various pathologies, including those involving the central nervous system, cardiovascular system, the respiratory system, and the gastrointestinal tract, among others. These aspects have been the focus of previous reviews [8,20,21,22,23,24,25]. Thus, this review is intended to primarily describe the progress made in unraveling the role of Annexin A1 in the kidney, in health and disease, which should help identify venues of further research in this field. In addition, in light of several recent advances regarding the role of Annexin A1 in the cardiovascular system [20,21,22,23,24,25], it will emphasize the more recent discoveries of its role in this field.

## 2. Annexins in the Kidney

Early immunohistochemical studies, of normal rat kidney, showed a marked presence of Annexin A1 in epithelia of Bowman’s capsule, the macula densa, and medullary/papillary collecting ducts, but mild staining in the thick ascending limb of the loop of Henle, while other segments were largely devoid of staining [4,26] (Figure 1). Others reported four annexins (with apparent molecular masses of about 32, 35, 45 and 67 kDa and labelled as I, II, V, and VI) in isolated rat renal glomeruli and glomerular mesangial cells [27]. More recently, prominent Annexin A1 expression has been demonstrated for renal interstitial fibroblasts [28] and Annexin A1 is among other proteins in exosomes derived from apical, but not basolateral, plasma membrane of mouse collecting duct cells [29].

The discovery of annexins in the kidney, and their differential localization, led to studies focused on determination of its functional roles. Accordingly, mesangial cell cultures, constitutively expressing annexin 1, alone, were shown to incorporate ^32^P with (tyrosine) phosphorylation being upregulated (2–3 fold) upon exposure to epidermal growth factor, angiotensin II, arginine-vasopressin or endothelin I [27]. Authors suggested that annexin 1 tyrosine phosphorylation, by the aforementioned biologically active peptides, occurs via G-protein-linked receptors and that a plausible role exists for annexin 1 in the mitogenic effect of angiotensin II, arginine-vasopressin, and endothelin I in rat glomerular mesangial cells. A subsequent study investigated the role of annexin 1 in the response of mesangial cells to DEXA and/or interleukin (IL)-1β in relation to (14 kDa Group II) PLA2 activity (Figure 1). Mesangial cells annexin 1 remained unchanged despite marked increases in PLA2 activity and protein levels in response to IL-1β and DEXA. In addition, DEXA did not induce translocation of annexin 1 from the cytosol to cell membrane or its secretion. It was concluded that DEXA-induced inhibition of PLA2 in rat mesangial cells is due to the suppression of its gene expression rather than being mediated via annexin 1 [30]. However, utilizing annexin 1-deficient U937 cells (e.g., used in monocyte differentiation studies) and mice lacking *ANXA1* gene, a clear role for Annexin A1 in the inhibition of cytosolic PLA2 has been suggested [12,14].

More recently, the recognition of expression of Annexin A1 in cells of the juxtaglomerular apparatus prompted the determination of its role in relation to cyclooxygenase-2 (COX-2) in rat [31]. Juxtaglomerular apparatus is formed by the glomerular afferent arteriole and the distal convoluted tubule and regulates glomerular filtration rate, solute transport, and blood pressure. Results indicated that (a) Annexin A1 is extensively co-localized with COX-2 in macula densa, (b) furosemide (a high ceiling diuretic and COX2 inducer) increased the expression of both COX-2 and Annexin A1 (which were highly co-localized), thereby suggesting that Annexin A1 expression is regulated by Na^+^/K^+^/2Cl^−^ cotransporter, (c) Annexin A1 deficient mice displayed greater COX-2 positive cells in macula densa cells along with greater juxtaglomerular renin secretion (normalized to glomerular number), (d) cultured macula densa cells, transfected with full-length rat Annexin A1, displayed downregulation of mRNA for COX-2, (e) DEXA treatment induced Annexin A1 mRNA expression and protein secretion and suppressed COX-2 mRNA, and (f) cyclosporine H (0.1μM)-induced inhibition of Fpr1 attenuated the effect of DEXA on COX-2 suppression in the macula densa while the Fpr2/Fpr3 antagonist, WRW4, had no effect. Based on their collective observations, authors concluded that Annexin A1 inhibits COX-2 in macula densa, with consequent inhibition of PGE2 synthesis, thereby serving as an intrinsic modulator of juxtaglomerular apparatus [31]. Thus, aside from inhibition of PLA2, it appears that Annexin A1 also inhibits COX2 with subsequent downregulation of prostanoids.

In conclusion, given the pivotal role of the juxtaglomerular apparatus in the regulation of glomerular filtration, sodium reabsorption, and blood pressure control, unraveling of the physiological role of Annexin A1 in the kidney is of major relevance and significance for homeostasis. In addition, mesangial cells expression of Annexin A1 is of importance given the role of these cells in maintaining structural integrity of glomerular microvascular bed and mesangial matrix homeostasis as well as a myriad of responses to hemodynamic and immunologic injuries (e.g., proliferation, hypertrophy, matrix synthesis and degradation, production of cytokines and growth factor, oxidative stress and apoptosis) [32], aspects that require further investigation.

## 3. Role of Annexin A1 in Pathologies of the Kidney

### 3.1. Annexin A1 and Immune/Inflammatory Responses in Kidney Pathologies

The role of Annexin A1 in immune and inflammatory responses has been the focus of intense investigation and reviewed previously [8,10,15]. The following section describes aspects reported for the kidney and summarized in Figure 2 and Table 1.

The early recognition that Annexin A1 reduces leucocyte migration, in response to cytokines and chemokines in animal models of inflammation, led to the comparison of its temporal effects to those of DEXA in the lipopolysaccharide (LPS)-induced inflammatory response in rat [33]. Accordingly, LPS superfusion resulted in the adhesion of leukocytes to the endothelium followed by increased emigration from post-capillary venules (over 2 h), effects which were attenuated by either Annexin A1 or DEXA. The activity of myeloperoxidase (MPO) was increased in the kidney, lung, and ileum after 6-h of exposure to LPS, thereby suggestive of neutrophil accumulation in these tissues. DEXA reduced LPS-induced elevation in MPO activity in the ileum but not in the lungs or kidneys, thereby suggesting differential responses of vascular beds in the ileum than those of the lung or the kidney. However, Annexin A1 did not affect LPS-induced increase in MPO activity in any of the aforementioned tissues. Authors concluded that Annexin A1 reduces leukocyte adhesion and emigration following a short-term (2 h), but not a longer-term (6 h) exposure to LPS. In this context, it is noteworthy that a mechanistic link between Annexin A1 and hydrogen sulfide pathway has been reported in leukocyte trafficking in experimental models of inflammation (e.g., induced by LPS) [34].

The potential role of Annexin A1 in relation to pathogenesis of the autoimmune disease of lupus nephritis (LN) has been investigated using renal biopsy and blood samples of patients afflicted with the disease [35]. Accordingly, microdissected glomeruli from biopsy samples of patients with LN were subjected to proteomic analyses. In addition, serum samples from large cohorts of patients with systemic lupus erythematosus, with and without LN, and glomerulonephritis of other etiologies were tested. Glomerular IgGs recognized 11 podocyte antigens although reactivity varied depending on LN pathology. Interestingly, IgG2 autoantibodies against Annexin A1, and α-enolase, were predominant and identified in 10 and 11 of the 20 biopsy samples, respectively. α-enolase is a metabolic enzyme but also serves multiple other functions (e.g., cell surface receptor for plasminogen). Immunohistochemical analyses revealed co-localization of Annexin A1 or α-enolase with IgG2 in glomeruli. In addition, high levels of serum anti-α-enolase IgG2 and/or anti-Annexin A1 IgG2 were detected in most patients with LN but not patients with other forms of glomerulonephritis. Serum levels of both autoantibodies decreased significantly after a year of LN therapy. Anti-α-enolase IgG2 recognized specific epitopes of α-enolase and did not cross-react with dsDNA. Although circulating dsDNA antibodies (which can interact with antigens on cell membranes of podocytes and mesangial cells) have been considered to play an important pathogenic role in LN, more recent information indicates that anti-DNA deposits account for 10–20% of IgG, thereby adding further relevance to findings of this study. Furthermore, monoclonal IgG2 derived from lupus-prone MRL-*lpr/lpr* mice recognized human α-enolase, suggesting homology between animal models and human LN. It was concluded that LN manifests a multi-antibody composition and that IgG2 autoantibodies against α-enolase and Annexin A1 predominate in the glomerulus and can also be detected in serum. Thus, given the increased predictive power of a combination of markers, the development of a panel composed of multiple antibodies (e.g., targeting Annexin A1, α-enolase, etc.) could serve as a useful surrogate biomarker of LN.

More recently, the role of Annexin A1, in the context of its cell death/survival function, has been reported in nephropathy associated with the autoimmune disease of glomerulonephritis with polyangiitis (GPA) [36]. This condition is associated with antinuclear-cytoplasmic antibodies (ANCA) targeting proteinase-3, thereby causing kidney injury. Indeed, increased neutrophil membrane proteinase-3 is a feature of apoptosis in this condition. Accordingly, proteomic analyses were carried out for comparison of neutrophils of patients with GPA and those of healthy controls under baseline condition and during apoptosis. It was reported that neutrophils proteome was markedly different at the onset of GPA and that this alteration became more pronounced following ex vivo apoptosis. Alterations in cell death/survival proteins were noted in neutrophils of patients with GPA including proteinase-3 binding proteins such as Annexin A1 (as well as calreticulin, and phospholipid scramblase 1). Indeed, increased membrane expression of Annexin A1 was a feature of apoptosis. It was proposed that the expression of these proteins at the cell membrane of apoptotic neutrophils, of patients with GPA but not healthy subjects, account for persistent inflammation and autoimmunity. Thus, differential expression of these proteins offers the opportunity for novel biomarker discovery and therapeutic option(s) for the management of kidney injury in GPA.

In conclusion, the immune system contributes to kidney damage that can be of long-term sequela and progression to chronic kidney disease. Autoantibodies against renal antigens (e.g., collagen IV in glomerular basement membrane) account for direct immune-mediated renal injury while immune complex formation, in systemic forms of autoimmunity, contribute to indirect immune-mediated kidney injury. The ensuing dysregulated inflammatory responses cause exacerbated recruitment of immune cells to the site of injury, which contribute to the pathogenesis of chronic kidney disease and, if left uninterrupted, to end-stage renal failure [37]. Given the multifaceted role of Annexin A1 in various inflammatory pathologies, the availability of its peptide and small molecule modulators [38] paves the way for establishing pathogenic mechanisms and potential therapeutic options for these conditions.

### 3.2. Annexin A1 and Acute Kidney Injury

An early study reported marked increase in Annexin A1 level in homogenates of the whole kidney two days after an ischemic insult. Further, immunohistochemical analyses showed restricted expression in some segments similar to non-ischemic kidneys (as alluded to earlier) but the thick ascending limb stained heavily following an ischemic insult suggestive of altered expression of Annexin A1 as a pathophysiological response [26]. Thereafter, the impact of an Annexin A1 mimetic peptide (Ac2-26) was explored in rat kidneys subjected to ischemia-reperfusion injury- a model of acute kidney injury (AKI) [39]. Administration of Ac2-26 before clamping of the renal artery conferred marked renoprotection as indexed by restoration of glomerular filtration rate and urine osmolality and prevented acute tubular necrosis. Further, the treatment reduced neutrophil extravasation and macrophage infiltration but did not influence lymphocyte movement. Interestingly, the ischemia-reperfusion insult increased Annexin A1 expression in renal epithelial cells, an effect that was attenuated by Ac2-26 administration. In addition, isolated proximal tubules suspension displayed reduced hypoxic/reoxygenation injury in response to Ac2-26. Authors concluded that while neutrophils are important mediators of renal injury in response to an ischemia-reperfusion insult, Annexin A1 is an important regulator of the defense response of epithelial cells in this condition and it may serve as a novel therapeutic option for relevant clinical conditions such as renal transplantation [39].

More recently, a rather detailed study examined the impact of an Annexin A1 tripeptide mimetic (ANXA1sp) on the outcome of AKI in the context of assessment of sirtuin-3 (SIRT-3) and mitochondrial function [40]. The focus on SIRT-3 relates to its ability to regulate pivotal cellular processes, including mitochondrial function and bioenergetics, mitophagy, and oxidative stress. Accordingly, utilizing the murine model of AKI, induced by renal ischemia reperfusion injury, ANXA1sp was administered an hour before renal vessel clamping and an hour after restoration of renal blood flow. Treatment with ANXA1sp reduced kidney damage as indexed by histologic injury score and serum creatinine level. In addition, it reduced cell death (TUNEL staining) in association with inhibition of oxidative stress and promotion of mitochondrial integrity and function (indexed by assessment of superoxide dismutase, 8-hydroxydeoxyguanine, oxyguanosine DNA glycosylase, mitochondrial transcription factor A) in the ischemic-reperfused kidneys. Similarly, ANXA1sp reduced cell death in HK-2 cells subjected to hypoxia. The impact of ANXA1sp on other markers of mitochondrial integrity and function were as follows. While ANXA1sp reduced dynamin-related protein-1 level (marker of mitochondrial fission) in ischemic-reperfused kidneys, suggestive of curtailing mitochondrial fragmentation, the treatment improved markers of mitophagy (e.g., LC3II, PTEN induced putative kinase 1 (PINK1), and Parkin) under these conditions. Importantly, ANXA1sp increased mRNA and protein expressions of SIRT3 in ischemic-reperfused kidneys compared to their vehicle-treated counterparts (but these parameters remained lower than those of their sham-operated controls). Further, in vitro studies showed that SIRT3 silencing reverses protection against hypoxic cell death conferred by ANXA1sp. Authors concluded that renoprotective effects of ANXA1sp relate to upregulation of SIRT3 and that it offers therapeutic potential for limiting kidney injury as a perioperative agent in conditions associated with renal ischemia (e.g., surgery and transplantation) [40].

In another study, serum Annexin A1 was used as an index of success of continuous renal replacement therapy (CRRT) in the setting of pyemic secondary acute kidney injury (AKI). Annexin A1 increased in this condition and its peak level in the CRRT-responder group decreased more rapidly compared to the non-responder group. Interestingly, however, other indices of kidney function (e.g., creatinine, urea and nitrogen levels) did not distinguish responders vs. non-responders to CRRT. Authors concluded that serum Annexin A1 level predicts the outcome of CRRT on pyemic secondary AKI [41].

In conclusion, AKI remains a major clinical problem as it affects critically ill patients and has detrimental effects on both short- and long-term outcomes. AKI, secondary to ischemia-reperfusion injury, is associated with several clinical conditions, including renal transplantation, partial nephrectomy, cardiac surgery, shock, and repair of some abdominal aneurysms. Multiple and diverse pathogenic mechanisms contribute to AKI including dysregulation of inflammatory mechanisms as both triggering and essential amplifying factors leading to an excessive inflammatory response thereby exacerbating tissue injury. Given the role of Annexin A1 in regulating multiple components of the immune and inflammatory responses, further investigation of its potential beneficial effects in such diverse conditions is warranted. Moreover, investigation of the impact of Annexin A1, or its mimetics, i.e., peptides and small molecules, initiated at various time points after the restoration of renal blood flow and determination of short- and long-term outcomes represent fertile ground for future investigation.

### 3.3. Annexin A1 and Immunosuppressive Agents

Despite the revolutionary impact of immunosuppressive agents on organ transplantation, nephrotoxicity (e.g., impaired renal hemodynamics) is a major adverse outcome associated with use of agents such as cyclosporine or tacrolimus. The administration of tacrolimus to rats impaired renal function manifested as increased renal vascular resistance and reductions in renal blood flow and glomerular filtration rate associated with increased macrophage infiltration and tubulointerstitial injury; these effects were accompanied with increased renal tissue expression of Annexin A1. Further, the administration of Ac2-26 attenuated tacrolimus-induced dilatation of renal tubules and macrophage infiltration, effects that could be of therapeutic value [42]. Thereafter, this group explored the impact of Ac2-26 in cyclosporine-treated rats [43]. Accordingly, they showed that cyclosporine treatment causes significant impairment of renal blood flow and glomerular filtration rate along with renal tubule dilatation and macrophage infiltration accompanied with increased kidney expression of Annexin A1. Treatment with Ac2-26 ameliorated renal hemodynamic changes, tubular damage, and macrophage infiltration in cyclosporine-treated animals. Despite these promising observations, their relevance to kidney dysfunction/injury in human subjects receiving cyclosporine or tacrolimus therapy (e.g., organ transplantation or autoimmune disorders) remains to be established.

### 3.4. Annexin A1 and Renal Injury of Diverse Etiopathogenesis

The observation of upregulation of Annexin A1 in kidney injury models (e.g., AKI and cyclosporine- and adriamycin-induced nephrotoxicity) has led to the determination of whether urinary level of Annexin A1 can serve as an index of glomerular injury in adriamycin-induced nephrotoxicity or for a heterogeneous groups of patients manifesting minimal change disease (MCD) vs. those with other forms of kidney pathologies (i.e., IgA nephropathy, membranous glomerulonephritis, focal segmental sclerosis, diabetic nephropathy, lupus nephritis, and crescentric glomerulonephritis) [44]. MCD refers to glomerular injury detected via electron microscopy and is the etiology of nephrotic syndrome in 10–25% of human subjects. It was reported that (a) urinary excretion of Annexin A1 is detectable in both the animal model of adriamycin-induced nephrotoxicity and in patients except those with MCD, (b) urinary Annexin A1 was positively correlated with various types of glomerular lesions, and (c) urinary Annexin A1 was detected in diabetic patients with otherwise normal albumin excretion. Authors concluded that Annexin A1 may serve as an early diagnostic and prognostic marker of glomerular injury and may differentiate patients with MCD vs. those with other pathologies of glomeruli [44].

### 3.5. Annexin A1 and Renal Fibrosis

Given the anti-inflammatory effects of Annexin A1, and its interaction with Fpr2, in protection against organ fibrosis, the role of Annexin A1 in kidney fibroblasts has been examined utilizing an animal model of renal fibrosis induced by inhibiting angiotensin II type 1 receptor with L-158.809 during the nephrogenic period [28]. The treatment apparently causes late-onset hypertensive nephropathy and fibrosis. In addition, effects of Annexin A1 were examined in cultured renal fibroblasts obtained from Annexin A1 knockout and wild type mice. The renal fibrotic model displayed matrix foci containing CD73^+^ fibroblasts, alpha-smooth muscle actin (α-SMA^+^) myofibroblasts and CD68^+^ macrophages. Further, in these animals, transforming growth factor-β (TGF-β) and Annexin A1 mRNAs were increased compared to those of controls. Annexin A1 expression was identified in macrophages and fibroblasts (with intense staining); however, myofibroblasts were negative. On the other hand, Fpr2 was expressed by fibroblasts, myofibroblasts, macrophages and endothelial cells. Further, Annexin A1 and Fpr2 immunoreactivity were increased in fibrotic foci, with fibroblasts and macrophages expressing both proteins. Renal fibroblasts obtained from Annexin A1 knockout animals displayed markedly higher α-SMA and collagen 1A1 mRNA levels than controls, with the effect much more pronounced for collagen 1A1 mRNA. Treatment of renal fibroblasts, obtained from wild type mice, with TGF-β increased mRNA levels of α-SMA and collagen 1A1. These changes were markedly attenuated in response to Annexin A1 overexpression. Authors concluded that ubiquitous expression of Annexin A1 and Fpr2 in the kidney interstitium regulates functional phenotype of renal fibroblasts and their contribution to synthesis of extracellular matrix proteins. Thus, it is likely that activation of annexin A1/Fpr2 signaling in fibroblasts could exert antifibrotic impact in chronic kidney disease. In this context, it is noteworthy that Annexin A1 gene expression is upregulated in renal cortex of the deoxycorticosterone-acetate/NaCl rat model of hypertension [45], a condition associated with marked kidney inflammation, fibrosis and dysfunction; increased Annexin A1 expression likely serves as a compensatory mechanism curtailing fibrosis.

### 3.6. Annexin A1 and Metabolic Diseases

Microvascular complications involving major target organs (e.g., kidney and heart) are common in diabetes mellitus for which inflammatory mechanisms play major pathogenic roles. Thus, the role of Annexin A1, as an endogenous anti-inflammatory mediator, has been examined in cardiorenal complications of type 1 diabetic patients and the streptozotocin (STZ)-induced diabetic mouse model [46]. Patients with type 1 diabetes, irrespective of nephropathy, displayed increased plasma Annexin A1. However, Annexin A1 level was reduced in renal and cardiac tissue lysates of diabetic animals. Further, Annexin A1 deficiency was associated with more severe STZ-induced diabetes accompanied with more severe nephropathy (indexed by albumin-to-creatinine ratio, glomerular area and fibrosis) and cardiomyopathy (indexed by ejection fraction, fractional shortening, etc.). To provide mechanistic linkage, it was shown that Annexin A1 knockdown in non-diabetic mice significantly increases phosphorylation of several MAPK (i.e., p38, ERK and JNK) in the heart and kidney, an effect that further increased with co-existence of STZ-induced type 1 diabetes. While prophylactic treatment of type 1 diabetic mice with human recombinant Annexin A1 (hrANXA1) reduced kidney and heart abnormalities, treatment after development of significant cardiorenal abnormalities in diabetic mice prevented further decline in cardiac and renal functions. In addition, treatment with hrANXA1 reduced the elevation in phosphorylation of MAPK and restored phosphorylation status of Akt in diabetic mice. It was concluded that increased plasma Annexin A1 occurs in type 1 diabetic subjects independent of significant impairment of kidney function. In addition, hrANXA1 protected against cardiorenal abnormalities of the murine model of type 1 diabetes via inhibition of MAPK but restoration of pro-survival activity of Akt [46]. Thus, Annexin A1 may represent a novel therapeutic option for microvascular complications of diabetes mellitus.

With respect to diabetes mellitus, increased renal Annexin A1 is shown to correlate with kidney function and outcomes in established diabetic nephropathy in a human cohort [47]. According, using the STZ-injected/high fat-fed (STZ/HF) mice, it was shown that Annexin A1 deficiency exacerbated renal injury manifested as exacerbated albuminuria, mesangial matrix expansion, tubulointerstitial lesions, kidney inflammation and fibrosis. On the other hand, Annexin A1 overexpression attenuated renal injury in diabetic mice. Further, treatment with Ac2-26 attenuated kidney injury in db/db mice (a genetic model of type 2 diabetes) and in diabetic Annexin A1 knockout mice. Mechanistic studies revealed that intracellular Annexin A1 binds to the p65 subunit of nuclear factor-κB (NF-κB), thereby inhibiting its activity and reducing inflammation [47]. Thus, Annexin A1 may serve as a therapeutic option to ameliorate diabetic nephropathy.

The role of Annexin A1 in diabetic nephropathy has been further investigated in the context of assessment of associated metabolic abnormalities [48]. Deletion of Annexin A1, in STZ/HF mice, exacerbated renal injury indexed by worsening of albuminuria, mesangial matrix expansion and tubulointerstitial injury. Further, Annexin A1 deficiency increased intra-renal accumulation of lipids and mitochondrial abnormalities. Treatment with Ac2-26 exerted beneficial effects against lipid accumulation in diabetic mice. Silencing of *ANXA1* in high-glucose/palmitic acid (HGPA)-treated proximal tubule epithelial cells (PTECs) suppressed phosphorylated Thr^172^ adenosine 5′-monophosphate-activated protein kinase, culminating in decreased expression of proliferator-activated receptor-α and carnitine palmitoyltransferase 1b accompanied with increased HGPA-induced lipid accumulation, apoptosis and increased expression of pro-inflammatory and pro-fibrotic genes. Further, renal biopsies of diabetic nephropathy patients displayed more intense staining for adipose differentiation-related protein (ADRP) than healthy controls with a negative correlation between ADRP and estimated glomerular filtration rate. Interestingly, tubulointerstitial Annexin A1 expression correlated with ADRP suggestive of its association with lipid abnormalities of diabetic nephropathy patients. It was concluded that Annexin A1 regulates lipid metabolism (e.g., of PTECs) to ameliorate disease progression, and that its modulation likely offering therapeutic potential for diabetic nephropathy.

In conclusion, dysregulation of Annexin A1 is involved in renal conditions of diverse etiopathogenesis. Interestingly, and as alluded to above, increased kidney expression of Annexin A1 is a feature of some of these conditions likely suggestive of serving as compensatory mechanism to curtail tissue injury (Table 1). Importantly, however, Annexin A1 can be cleaved, at the N-terminal domain, by a number of enzymes including elastase, proteinase 3, metalloproteinases, cysteine protease, calpain I or aspartyl protease cathepsin D among others [7]. Interestingly, products of some of these biotransformation pathways (e.g., by calpain or cathepsin D) are believed to exert pro-inflammatory effects thereby interfering with Annexin A1′s anti-inflammatory and pro-resolving activity [7]. This likely explains why compensatory increase in Annexin A1 in some of the aforementioned conditions may be insufficient to curtail inflammation while treatment with Annexin A1 mimetics restores tissue homeostasis. Nonetheless, while Annexin A1 mimetic peptides (e.g., Ac2-26) serve as valuable experimental tools, the development of orally bioavailable agonists of Annexin A1 receptors (i.e., Fpr1/2) would help better establish therapeutic values of modulation of Annexin A1 signaling.

### 3.7. Annexin A1 and Kidney Stone

Elevation in urinary calcium level is associated with impaired kidney function and kidney stone formation. Thus, proteomic analysis of renal tubular cells was carried to determine whether their protein profile is altered upon exposure to a high calcium medium [49]. Accordingly, it was shown that Madin Darby Canine Kidney (MDCK) cells exposed to high calcium displayed increased capacity for binding to calcium oxalate monohydrate crystals accompanied with increased Annexin A1 expression, as a calcium-binding protein, on the apical surface of the cells, an effect reversed by pre-treatment with antibody targeting Annexin A1. Importantly, exposure to high calcium also led to impaired wound healing and cell proliferation. It was concluded that these in vitro observations implicate a role for Annexin A1 in nephrolithiasis and kidney dysfunction that may be of relevance to hypercalciuria and kidney stone formation.

A subsequent study further explored the role of Annexin A1 in kidney stone formation in the context of an assessment of the effects of caffeine in this process given reports that caffeinated beverages reduce the risk of kidney stone formation [50]. Caffeine reduced calcium oxalate monohydrate crystal numbers but increased crystal size thereby resulting in no change in the size of the crystal mass. Interestingly, however, caffeine treatment, dose-dependently, reduced the ability of MDCK cells to bind crystals, an effect associated with decreased expression of Annexin A1 on the apical surface suggestive of its translocation into the cytoplasm; no change was reported for several other calcium oxalate monohydrate crystal binding proteins (i.e., annexin-A2, α-enolase, HSP70, and HSP90). Further, caffeine reduced intracellular Ca^2+^ level but increased the calcium secretory index. Authors concluded that caffeine exerts a protective effect against kidney stone formation likely via the translocation of Annexin A1 from the apical membrane of cells into the cytosolic compartment, thereby reducing the crystal-binding capacity of renal tubular epithelial cells. In a more recent study, these investigators explored the role of Annexin A1 in the purported protective effect of estrogen on kidney stone formation given reports of lower risk in female than male [51]. Utilizing MDCK cells coupled with proteomic analyses, authors reported that seven-day exposure to estrogen resulted in the differential expression of 58 proteins which are primarily involved in functional networks related to “binding and receptor,” “metabolic process,” and “migration and healing”. Further, estrogen-treated cells displayed reduced calcium oxalate crystal-binding capacity coupled with decreased cell membrane levels of Annexin A1 and α-enolase (as calcium oxalate crystal-binding receptors). Interestingly, estrogen reversed the elevations in cell surface expression of Annexin A1, and α-enolase, secondary to high-calcium and high-oxalate challenge. Estrogen also decreased intracellular ATP level and enhanced cell migration and tissue healing. Thus, it is likely that estrogen promotes cellular processes that favor reduced cell surface expression of receptors for calcium oxalate crystals and enhanced cell proliferation and tissue healing, thereby contributing to prevention of stone formation.

In conclusion, nephrolithiasis is often a painful condition with its prevalence increasing worldwide. The annual incidence is estimated to be about eight per 1000 adults that peaks about midlife in developed countries. Most cases of kidney stones are of noninfectious etiologies but are associated with environmental factors (climate and fluid intake) and several comorbidities including metabolic syndrome/obesity, excessive protein intake and gout. Shock wave therapy and surgery are among management approaches but patients experience recurrent stones within 10 years after removal [50]. Thus, aside from preventive measures (e.g., hydration), attention is increasingly focused on modulation of crystal formation, growth and aggregation to decrease or prevent their adhesion to renal epithelial cells. The studies focused on the role of Annexin A1 in pathogenesis of nephrolithiasis suggest that downregulation of renal cell membrane Annexin A1 may offer a promising approach. Thus, the development of therapeutically useful inhibitors of Annexin A1 is a step in that direction.

### 3.8. Annexin A1 and Renal Cancer

The role of Annexin A1 has been extensively studied in a number of cancers and tumors cell lines [52]. With respect to kidney cancer, early recognition of Annexin A1-mediated growth inhibition of glucocorticoids in human lung adenocarcinoma cell line led to the determination of the impact of hydrocortisone and hrANXA1 on cultured human mesangial cells (CHMC) [53]. Annexin A1 dose-dependently inhibited proliferation of CHMC (unstimulated or activated with platelet-derived growth factor) as assessed by ^3^H-thymidine incorporation and cell count. Similarly, hydrocortisone dose-dependently inhibited CHMC proliferation. Treatment with an antibody against Annexin A1 reversed hydrocortisone-induced inhibition of CHMC proliferation. Further studies using flow cytometry and indirect immunofluorescent microscopy indicated specific Annexin A1 binding sites on the surface of CHMC. It was concluded that corticosteroids cause generation/secretion of Annexin A1 by CHMC that, in turn, results in suppression of their proliferation via specific binding sites on CHMC.

The role of Annexin A1, in conventional renal cell carcinoma (CRCC), without or with metastasis, was investigated with respect to tumorigenesis, metastatic potential and clinical outcome [54]. Immunohistochemical staining was correlated with Fuhrman nuclear grade (an independent predictor of cancer-specific survival), amount of eosinophilic cells and higher clinical stage leading to the conclusion that that Annexin A1 could serve as a prognostic marker for tumor progression. Another study assessed cytoplasmic staining of Annexin A1 in metastatic renal cell carcinoma (mRCC; 77 cases) treated with nephrectomy followed by therapy with sunitinib (a small molecule inhibitor of receptor tyrosine kinase) [55]. Annexin A1 negative cases (32%) showed better progression-free survival and overall survival than those cases with positive cytoplasmic Annexin A1 staining. It was concluded that cytoplasmic expression of Annexin A1 was a negative predictive marker for sunitinib treatment in mRCC patients, likely because its anti-angiogenic effect cannot overcome the pro-angiogenic effect of Annexin A1.

More recently, assessment of Annexin A1 expression in renal cell carcinoma of 27 patients was coupled with the use of log-rank test for assessment of disease-free survival [56]. Accordingly, 48% of cases displayed high Annexin A1 expression in cell membranes of tumor cells while others showed weak or negligible staining. Further, patients whose tumors stained intensely for Annexin A1 had significantly poorer disease-free survival than others. Knockdown of Annexin A1 in Caki-2 cells (a human clear cell renal cell carcinoma line) resulted in reduced proliferation, invasion, motility and adhesion compared to control cells. Further, Annexin A1 deficient cells expressed lower relative levels of membrane-type 1 matrix metalloproteinase and hypoxia-inducible factor 1-α transcripts. It was concluded that Annexin A1 is associated with the malignant potential of renal cell carcinoma and, thus, could serve as a marker of poor prognosis.

In addition to context-specific tumorigenesis, many other functions have been attributed to Annexin A1 including clearance of apoptotic cells [57]. Accordingly, expression of Annexin A1 was assessed in fetal kidneys (at different gestational periods), mature kidneys and in kidney cancer tissues to gain insight into its localization and its potential role during nephrogenesis and in renal tumors. Annexin A1 was expressed in mesangial cells and podocytes of developing glomeruli and renal cortex of fetal kidneys (days 14–19 of gestation); mesangial cells Annexin A1 expression declined over time but persisted into adulthood. Importantly, Annexin A1 expression increased with the progression of clear cell renal cell carcinoma (and other cancer types), suggestive of its potential role in tumorigenesis. It was concluded that Annexin A1 plays important functional roles in podocytes and mesangial cells, as integral components of renal glomeruli, and that their unraveling could help with the development of diagnostic, prognostic, and therapeutic strategies for various renal pathologies.

In conclusion, Annexin A1 has been implicated in cell proliferation, apoptosis, sensitivity to chemotherapy as well as invasion and metastasis, aspects that are essential in cancer biology. Nonetheless, it is suggested that Annexin A1 may serve as a partial functional mediator of tumorigenesis and metastasis rather than a tissue-specific mediator for predicting cancer initiation and/or metastasis [52]. With respect to renal cancer, and as described above, while the inhibitory effect of Annexin A1 on the proliferation of mesangial cells has been reported, other studies suggest tumorigenesis and malignant potentials. Thus, the development and use of effective inhibitors of Annexin A1 signaling would help better understand the role of Annexin A1 in tumor biology.

## 4. Recent Developments on the Role of Annexin A1 in the Cardiovascular System

Several recent reviews have described the role of Annexin A1 in the cardiovascular system, including its pro-resolving role in inflammation associated with atherosclerosis and myocardial infarction [20,21,22,23]. Results of some of these studies are summarized in Table 2 and Table 3 [58,59,60,61,62,63,64,65,66,67]. Thus, the following section provides a review of the more recent studies in this field (Table 4).

Microcalcification promotes cardiovascular disease and its pathogenic mechanisms include contributions from extracellular vesicles (EVs), which include plasma membrane-derived microvesicles, and endosomal-derived exosomes. To explore these aspects, human coronary artery specimens, human aortic valve tissue and primary human coronary smooth muscle cells were utilized coupled with proteomics and the use of single-EV microarray to distinguish microvesicles from exosomes [68]. Increased Annexin A1 was present primarily on aggregating and calcifying microvesicles; Annexin A1 vesicle aggregation was suppressed in response to calcium chelation and neutralizing antibody targeting Annexin A1. Further, knockdown of Annexin A1 reduced EV aggregation and microcalcification in human cardiovascular cells and acellular 3D-collagen hydrogels. Thus, it is likely that Annexin A1 is involved in formation of microcalcification in vulnerable regions of plaque, an aspect of relevance and importance for EV-associated diseases of the cardiovascular system, autoimmune and neurodegenerative diseases [68].

Acute aortic dissection (AAD) is a life-threatening disease associated with high morbidity and mortality with a hallmark feature of vascular smooth muscle cells (VSMCs) phenotype switching. Immunohistochemical studies showed marked Annexin A1 expression in mouse AAD tissues and micro-CT studies revealed a role for Annexin A1 in the development of AAD, as exemplified by its deficiency causing progression of the disease [69]. On the other hand, treatment with Ac2-26 rescued the AAD phenotype in Annexin A1 knockout mice. Annexin A1′s role in VSMCs phenotype switching was revealed by the observation that its deficiency triggered the synthetic phenotype of VSMCs accompanied with increased inflammation and matrix metalloproteinases (MMP) production and augmented degradation of elastin. Deficiency of Annexin A1, restricted to VSMCs in mice, recapitulated VSMCs phenotype switching associated with the exacerbation of AAD. Importantly, key findings of the murine model were observed in aortic tissue specimens of AAD patient, particularly the observation that reduced Annexin A1 expression is associated with VSMCs phenotype switch, exacerbated inflammation and enhanced MMP production in human aorta. It was concluded that modulation of Annexin A1 signaling may offer a therapeutic option for AAD.

The role of Annexin A1 has been investigated in microvascular complications of metabolic syndrome [70]. Deficiency of Annexin A1, in high fat-fed mice, caused more severe abnormalities as revealed by assessment of insulin resistance, dyslipidemia, hepatosteatosis and urinary protein excretion compared to their wild type counterparts. Further, treatment of high fat-fed wild type mice with hrANXA1 reduced the aforementioned abnormalities. Authors also reported that Annexin A1 knockout mice exhibit constitutively activated Ras homolog family member A (RhoA) and that the diabetic mice with reduced Annexin A1 also displayed activated RhoA. Interestingly, treatment of high fat fed-mice with hrANXA1restored tissue levels of Annexin A1 associated with inhibition of RhoA activity and consequent restoration of the activities of Akt, glycogen synthase kinase-3β and endothelial nitric oxide synthase, effects attributed to re-sensitization of insulin receptor substrate (IRS)-1 signaling. Further, in human hepatocytes exposed to hyperglycemia, Annexin A1 protected against excessive mitochondrial proton leak via activating Fpr2. It was concluded that Annexin A1 is a pivotal regulator of RhoA activity, restoring IRS-1 signaling, and that hrANXA1 may offer therapeutic benefit for complications of type 2 diabetes mellitus.

The introduction of more stable small-molecule Fpr1/2 agonists, Compound 17b and Compound 43, has led to the investigation of their effects [38]. While both compounds exert similar effects, Compound 17b seemingly exerts differential effects on intracellular calcium metabolism likely accounting for its greater cardioprotective effects than compound 43. Authors reported predominant localization of Fpr1 and Fpr2 on murine aortic vascular smooth muscles. Further, Compound 17b, but not Compound 43, caused concentration-dependent but endothelium-independent relaxation of mouse aorta. Further, Compound 17 abrogated concentration-dependent CaCl_2_-induced contraction of aorta, primed with increased potassium, an effect attributed to inhibition of calcium mobilization via voltage-gated calcium channels. Importantly, chronic treatment of STZ-induced diabetic mice with Compound 17b reversed endothelial dysfunction of aortae via increased generation of vasodilator prostanoids. It was concluded that as an endothelium-independent vasodilator, Compound 17b exerts vasoprotective effects in diabetes mellitus.

The anti-apoptotic effect of Ac2-26 in relation to cardiomyocyte injury has been explored utilizing the cecal ligation and puncture model of septic shock coupled with in vitro studies using H9C2 cells exposed to lipopolysaccharide (LPS) [71]. Accordingly, treatment with Ac2-26 prevented myocardial ultrastructure damage and reduced apoptosis of myocardial cells. Similarly, Ac2-26 reduced LPS-induced apoptotic cell death in H9C2 cells associated with a significant upregulation of lipoxin A4 (LXA4) receptor (i.e., Fpr2/ALX) but downregulation of PI3K and Akt. Further, Ac2-26 treatment decreased NF-κB activity and tumor necrosis factor-α level. Ac2-26 treatment also suppressed the activities of caspase-3/8 in H9C2 cells. It was concluded that Ac2-26 curtails sepsis-induced cardiomyocyte apoptosis via LXA4/PI3K/AKT signaling pathway.

With respect to myocardial infarction, the role of Annexin A1 in neutrophil infiltration and MPO activity were explored in rat [72]. Myocardial infarction reduced mRNA and protein expressions of Annexin A1, signal transducer and activator of transcription 3 (STAT3) and vascular endothelial growth factor (VEGF) as well as phospho-STAT3 level. Overexpression of Annexin A1 reduced levels of inflammatory factors, neutrophil infiltration and their apoptosis in association with improved cardiac functional indices. On the other hand, downregulation of Annexin A1 inhibited STAT3 activation and suppressed VEGF expression. It was concluded that Annexin A1 upregulation inhibits neutrophil infiltration and MPO activity likely via the activation of STAT3 signaling pathway in the rat model of myocardial infarction.

A recent study explored the therapeutic potential of hrANXA1 in heart failure with preserved ejection fraction (HFpEF) that can accompany rheumatoid arthritis [73]. Accordingly, hrANXA1 treatment of a murine model of rheumatoid arthritis, which also manifests diastolic dysfunction, not only prevented the progression of heart disease but also reversed established diastolic dysfunction in association with reduction in adverse cardiac remodeling. Indeed, the cardioprotection of hrANXA1 was associated with (a) reduced fibroblast populations and profibrotic markers (e.g., TGF-β, collagen 1A1, galectin-3), (b) modulation of cardiac immune cells including decreased activated T cell infiltration but increased MHC II^low^ macrophage (manifesting reduced pro-fibrotic activity) infiltration and (c) reduced cardiac proinflammatory cytokines (e.g., IL-6, IL-1β). Interestingly, however, the beneficial effects of hrANXA1 on myocardial diastolic dysfunction were not accompanied with marked modulation of severity of rheumatoid arthritis. Thus, authors proposed that inclusion of hrANXA1 (or an Fpr2 agonist), in therapeutic regimens, may represent a novel approach to treatment of HFpEF in inflammatory arthritis.

The potential impact of statin therapy on Annexin A1 was explored in patients with acute coronary syndrome (ACS) [74]. Serum Annexin A1 level was reduced in ACS patients than healthy controls. However, ACS patients receiving rosuvastatin displayed increased level of serum Annexin A1 than their counterparts not receiving the medication. It remains to be established whether other statins exert a similar effect in this condition.

With respect to use as a biomarker, a recent study concluded that elevated plasma Annexin A1 level is associated with worse congestion, higher risk for further elevation of creatinine, and higher rates of 60-day morbidity or all-cause mortality and 180-day all-cause mortality in hospitalized patients with acute heart failure associated with impaired kidney function (Table 1) [75].

**Table 1 cells-10-03420-t001:** Annexin A1 as a diagnostic/prognostic biomarker.

Condition	Finding	Citation(s)
Lupus Nephritis	Increased serum IgG2 against Annexin A1	[35]
Glomerulonephritis with polyangiitis	Increased Annexin A1 expression in neutrophils	[36]
Kidney injuries of diverse pathogenesis (e.g., diabetes mellitus, pyemic secondary AKI, Adriamycin-induced nephrotoxicity, etc.)	Increased kidney, serum (or plasma) and/or urinary Annexin A1	[41,42,43,44,45,46]
Patients with acute heart failure associated with impaired kidney function	Increased plasma Annexin A1 associated with worse congestion, morbidity and mortality	[75]
Renal cancer	Marker of poor prognosis Predictor of poor sunitinib response	[54,55,56,57]

**Table 2 cells-10-03420-t002:** Role of and therapeutic potential of Annexin A1 in atherosclerosis.

Model/Condition	Findings	Citation(s)
Human coronary atherosclerotic plaques	Expression of annexin-I in macrophages, with foam cell phenotype, in tunica intima and adventitia High expression of annexin-I in macrophages in plaque areas, containing TUNEL positive cells, is suggestive of phagocytosis of apoptotic cells	[58]
Patients with carotid artery stenosis (undergoing carotid endarterectomy)	Higher expression of Annexin A1 in carotid plaques of asymptomatic than symptomatic patients thereby suggestive of its protective role in atherosclerosis	[67]
Patients with or without recent acute cerebrovascular symptoms	Increased Annexin A1 expression in plaque-derived smooth muscle cells of the asymptomatic group	[59]
Administration of nanoparticles containing Ac2-26, with a lesion-targeting collagen-IV motif, to mice with established atherosclerosis	Decreased lesion size in association with reduced oxidative stress and necrotic area but increased plaque stability and lesional interleukin-10	[64]
Treatment of mice prone to develop atherosclerosis with hrANXA1	No effect on initiation of plaque formation but attenuated progression of existing plaques of aortic arch and subclavian artery	[65]
Administration of Ac2-26 to mice with early stage atherosclerosis ApoE^−/−^, Fpr2^−/−^ and ApoE^−/−^Anexxin-A1^−/−^ mice	Curtailed early atherogenesis in a receptor-dependent manner associated with decreased lesional size and macrophage accumulation Display increased atherosclerotic lesion size as well as myeloid cell adhesion and recruitment	[66]

**Table 3 cells-10-03420-t003:** Therapeutic potential of Annexin A1 modulation in cardiac injury models.

Model/Condition	Findings	Citation(s)
Effect Ac2-26 on sepsis-induced cardiomyocyte apoptosis	Attenuated cell death	[71]
Treatment of rats, subjected to acute myocardial ischemia-reperfusion injury (IRI), with hrANXA1	Reduced infarct size in association with reduced leukocyte extravasation, myeloperoxidase activity, TNF-α and macrophage inflammatory protein-1a	[60]
Treatment of rats, subjected to acute myocardial IRI, with Ac2-26	Reduced myeloperoxidase activity and interleukin-1β in the infarcted heart; protective effects were abrogated with an antagonist of Fpr receptors	[61]
Ac2-26 treatment of wild-type and Fpr null mice subjected to cardiac IRI	Cardioprotection in both wild-type and Fpr null mice	[62]
Effects of of Annexin A1(2-50) peptide in wild-type, Fpr1^−/−^, and Fpr2^−/−^/ALX^−/−^ mice. Effects of the metabolically stable form of this peptide (i.e., CR-Annexin A1(2-50) in the murine model of cardiac IRI	Reduced leukocyte adhesion in wild-type and Fpr1^−/−^, but not Fpr2^−/−^/ALX^−/−^, mice Reduced infarct size and incidence of 24-h death	[63]
Effects of hrANXA1 in ANXA1 deficient STZ mice	Ameliorated cardiac injury	[46]
ANXA1 overexpression in the rat model of myocardial infarction	Reduced infarct size and improved functional outcome in association with reduced levels of pro-inflammatory factors, neutrophil infiltration and their apoptosis	[72]

**Table 4 cells-10-03420-t004:** Role of and therapeutic potential of Annexin A1 in cardiovascular disorders.

Model/Condition	Findings	Citation(s)
Extracellular vesicles aggregates and microcalcification	Promotion by Annexin A1	[68]
Acute aortic dissection	Possible protection by promoting Annexin A1 signaling	[69]
Murine model of metabolic syndrome	Beneficial effects of hrANXA1 against metabolic and microvascular abnormalities	[70]
Vascular complication of metabolic syndrome	Beneficial impact of small molecule agonist of Annexin A1 receptors	[38]
Murine model of rheumatoid arthritis with diastolic dysfunction	Treatment with hrANXA1 prevented progression of heart disease and also reversed established diastolic dysfunction in association with curtailing cardiac inflammation and fibrosis.	[73]
Acute coronary syndrome patients	Rosuvastatin therapy increased serum Annexin A1 level	[74]

In conclusion, among cardiovascular disorders, considerable information is available regarding the role of Annexin A1 with respect to atherosclerosis and myocardial infarction where potentiating Annexin A1 signaling seemingly exerts beneficial outcomes (Table 2 and Table 3). Nonetheless, studies focused on therapeutic potential of Annexin A1 have utilized animal models of acute myocardial infarction. Those who survive the acute episode of myocardial infarction experience adverse cardiac remodeling and heart failure is an important sequel of the acute insult. Thus, studies focused on long-term outcomes of Annexin A1 modulation in cardiovascular disorders is necessary to better establish its therapeutic potential (Table 4). Further, the role of Annexin A1 in the most prevalent cardiovascular disorder, i.e., systemic hypertension, remains largely unexplored.

## 5. Perspective

Despite the well-documented anti-inflammatory effects of glucocorticoids, their use is associated with a number of serious adverse effects [76]. This recognition has led to intense research focused on unraveling the molecular mechanisms by which glucocorticoids regulate immune and inflammatory processes with the ultimate objective to devise novel therapies to avoid adverse effects of glucocorticoids. This has resulted in identification of glucocorticoid-inducible proteins including Annexin A1 and glucocorticoid-induced leucine zipper (GILZ). The renal and cardiovascular effects of GILZ as well as its therapeutic potential for associated diseases is the focus of a recent review [76]. This review complements that report as it focuses on the role of Annexin A1 in the kidney and cardiovascular system as well as the therapeutic potential of its modulation for associated disorders. As described above, and summarized in Figure 2 and Table 2, Table 3 and Table 4, it appears that while inhibition of Annexin A1 may offer therapeutic advantage in renal cancer and nephrolithiasis, for most of the other conditions potentiation/facilitation of Annexin A1 signaling may be advantageous. Thus, the development of orally bioavailable and effective inhibitors and activators of Annexin A1 should help to better establish the role of Annexin A1 in such diverse but significant clinical conditions. It is noteworthy that the assessment of the literature with respect to the effects of GILZ [76] and Annexin A1 on the kidney and the cardiovascular system indicates similarities among studies. For example, with respect to acute kidney injury simulated by ischemia reperfusion injury, treatment with either Annexin A1 peptide mimetics [39,40] or the cell-permeable GILZ protein [77] exert marked renoprotection. With respect to renoprotective effects of GILZ, decreased neutrophil flux and increased regulatory functional phenotype (i.e., N2) are suggested as major players [77]. Interestingly, however, while Annexin A1 is well known as a potent inhibitor of neutrophil transmigration, it remains to be established whether Annexin A1 also influences neutrophil polarization for conditions described in this submission although it is known to promote the anti-inflammatory phenotype of macrophages [20]. In this context, it is noteworthy that the impairment of neutrophil migration in GILZ knockout animals is attributed to a deficit in (DEXA-induced) Annexin A1 production [16]. Thus, while much has been learned from studies focused on renal and cardiovascular effects of Annexin A1 (and GILZ), major gaps remain, some of which have been alluded to in this communication, including potential cross-talk between Annexin A1 and GILZ in the regulation of neutrophil trafficking and function. The elucidation of these aspects offers fertile ground for future investigations to better establish the therapeutic potential of glucocorticoid-inducible proteins, Annexin A1 and GILZ.

## Figures and Tables

**Figure 1 cells-10-03420-f001:**
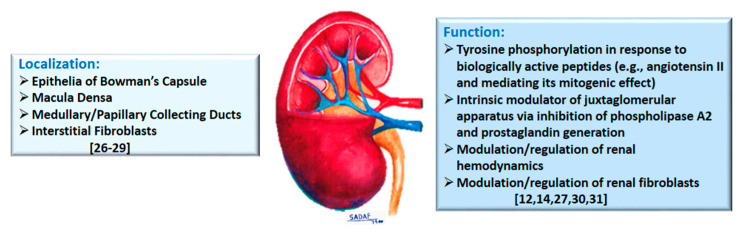
The diagram summarizes localization and physiological functions attributed to Annexin A1 in the kidney; relevant citations are provided in brackets.

**Figure 2 cells-10-03420-f002:**
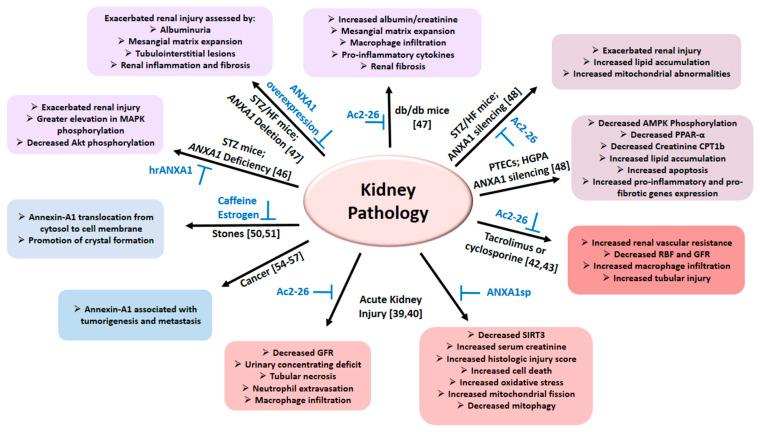
The diagram summarizes the role of Annexin A1 in various pathologies of the kidney as detailed in the text; relevant citations are provided in brackets. Abbreviations: AMPK: Adenosine 5′-monophosphate-activated protein kinase; ANXA1: Annexin A1; Ac2-26: Annexin A1 peptide mimetic; ANXA1sp: Annexin A1 tripeptide mimetic; CTP1b: Carnitine palmitoyltransferase 1b; GFR: Glomerular filtration rate; HFD: High fat diet; HGPA: High glucose palmitic acid; PPAR-α: Proliferator-activated receptor-α; PTECs: Proximal tubule epithelial cells; RBF: Renal blood flow; Sirtuin-3 (SIRT3); STZ: Streptozotocin.

## Data Availability

Not applicable.

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
