# Peer review of "Therapeutic Potential of Annexin A1 Modulation in Kidney and Cardiovascular Disorders"

_cells, 2021, doi:10.3390/cells10123420_

Round 1

Reviewer 1 Report

This elegant review addresses the role of annexin A1 (AnxA1) in the inflammatory response and tissue injury related to kidney and cardiovascular diseases, highlighting the therapeutic potential of AnxA1 (or its mimetic peptides) in these clinical conditions. This well-written manuscript provided an overview of the most recent and relevant data on this subject. Congratulations to the author. However, I would like to offer the following suggestions to improve it further:

1) It was difficult to read the figures’ content because of their low resolution. In addition to correcting the resolution, I suggest reducing the text within the diagrams (perhaps making them more illustrative). This would also allow increasing the font size, which would make it easier to read.

2) It is worth mentioning the main cellular and tissues sources of AnxA1, and its different affinities to each FPR receptor in the Introduction section.

3) Under inflammatory conditions, the intact form of AnxA1 (37 KDa) can be cleaved by proteases, such as neutrophil elastase, generating cleaved isoforms (e.g. 33 kDa), which are inactive or may have pro-inflammatory functions. Perhaps, in some pathological contexts in which AnxA1 expression is up-regulated, there is an increase in AnxA1 breakdown, possibly due to increased levels or activity of proteases, thus interfering with AnxA1 anti-inflammatory / pro-resolving activity. This could explain why the compensatory increase of AnxA1 in some inflammatory conditions may be insufficient to dampen inflammation, whereas treatment with AnxA1 (recombinant protein or peptides derived from the AnxA1 N-terminus) restores tissue homeostasis. This issue should be addressed in the Introduction or elsewhere, as appropriate.

4) Page 13 (line 587): Please referrer to the lipoxin A4 receptor as ALX/FPR2 to highlight that both AnxA1 and LXA4 bind to the same receptor.

Author Response

I am grateful to the reviewers for their kind remarks regarding this submission and for their very helpful/insightful comments and suggestion.  I have carefully considered all comments in revision of the manuscript.  The changes in response to reviewers’ comments are highlight in BLUE.

Reviewer 1:

This elegant review addresses the role of annexin A1 (AnxA1) in the inflammatory response and tissue injury related to kidney and cardiovascular diseases, highlighting the therapeutic potential of AnxA1 (or its mimetic peptides) in these clinical conditions. This well-written manuscript provided an overview of the most recent and relevant data on this subject. Congratulations to the author. However, I would like to offer the following suggestions to improve it further:

I am grateful to the reviewer for her/his kind remarks as well as the very valuable and insightful suggestions.

1) It was difficult to read the figures’ content because of their low resolution. In addition to correcting the resolution, I suggest reducing the text within the diagrams (perhaps making them more illustrative). This would also allow increasing the font size, which would make it easier to read.

I regret the poor resolution of figures.  In consideration of your comments and those of other reviewers, I now provide 2 Figures and 4 Tables summarizing findings of relevant studies along with citations.  I hope this format is more reader-friendly.  

2) It is worth mentioning the main cellular and tissues sources of AnxA1, and its different affinities to each FPR receptor in the Introduction section.

The Introduction is revised to incorporate reviewer’s suggestion (BLUE highlights).

3) Under inflammatory conditions, the intact form of AnxA1 (37 KDa) can be cleaved by proteases, such as neutrophil elastase, generating cleaved isoforms (e.g. 33 kDa), which are inactive or may have pro-inflammatory functions. Perhaps, in some pathological contexts in which AnxA1 expression is up-regulated, there is an increase in AnxA1 breakdown, possibly due to increased levels or activity of proteases, thus interfering with AnxA1 anti-inflammatory / pro-resolving activity. This could explain why the compensatory increase of AnxA1 in some inflammatory conditions may be insufficient to dampen inflammation, whereas treatment with AnxA1 (recombinant protein or peptides derived from the AnxA1 N-terminus) restores tissue homeostasis. This issue should be addressed in the Introduction or elsewhere, as appropriate.

I am truly grateful to the reviewer for pointing out this very important aspect that is now incorporated in the revised submission (Page 15. BLUE highlight) 

4) Page 13 (line 587): Please referrer to the lipoxin A4 receptor as ALX/FPR2 to highlight that both AnxA1 and LXA4 bind to the same receptor.

This aspect is now addressed in the Introduction and also in the segment indicated by the reviewer. 

Reviewer 2 Report

The author has synthetized recent work regarding the role of Annexin A1 and its therapeutic potential in kidney and cardiovascular diseases. The review brings together many interesting concepts and results. However, some paragraphs could be written in a more intelligible way. The author may want to address the following specific suggestions to further improve his manuscript.

1- The abstract could be reorganized to explain more clearly the double focus on kidney and cardiovascular disease.

2- My main concern relates to figures. Indeed, both figures mainly consist in long lists of effects, that do not provide much added value to the review in my opinion. In addition, they are barely legible due to very poor quality. The author should improve these figures by making them much more synthetic and integrated and/or replace them by figures illustrating biological effects of Annexin A1. A synthetic table or figure of the effects/therapeutic potential of AnnexinA1 focused on mechanisms rather than pathology would help.

4- The Line 183-184 “Thus, development of a panel composed of multiple antibodies could serve as novel surrogate biomarker of LN”. The author should clarify how the development of a panel of antibodies would be a biomarker or rephrase this sentence to make it clearer.

5- Throughout the text, they are several convoluted sentences that are difficult to understand as they stand (e.g. Line 198 “It was proposed that these proteins form a platform at the cell membrane of neutrophils, of patients with GPA but not healthy subjects, undergoing apoptosis and account for persistent inflammation and autoimmunity thereby offering the opportunity for novel biomarker discovery and therapeutic option(s) for management of kidney injury in GPA.”). I would suggest a re-reading of the article to make the sentences that require it more intelligible.

Minor points:

  • DEXA is defined twice
  • Throughout the text, the naming of Annexin 1/I should be homogenized
  • There are several typos (e.g. Line 96 “exits”, lines 155 ad 552 “anenxin”, line 205 “collage IV”…) that I did not list exhaustively. A careful reading of the review will allow for correcting these spelling errors.
  • Line 84 : “Others reported four annexins (with apparent molecular masses of about 32, 35 and 67 kDa” : only 3 apparent molecular masses are listed, is this a mistake ?

Author Response

I am grateful to the reviewers for their kind remarks regarding this submission and for their very helpful/insightful comments and suggestion.  I have carefully considered all comments in revision of the manuscript.  The changes in response to reviewers’ comments are highlight in BLUE.

Reviewer 2:

Comments and Suggestions for Authors

The author has synthetized recent work regarding the role of Annexin A1 and its therapeutic potential in kidney and cardiovascular diseases. The review brings together many interesting concepts and results. However, some paragraphs could be written in a more intelligible way. The author may want to address the following specific suggestions to further improve his manuscript.

1- The abstract could be reorganized to explain more clearly the double focus on kidney and cardiovascular disease.

This aspect is addressed in the Abstract. 

2- My main concern relates to figures. Indeed, both figures mainly consist in long lists of effects, that do not provide much added value to the review in my opinion. In addition, they are barely legible due to very poor quality. The author should improve these figures by making them much more synthetic and integrated and/or replace them by figures illustrating biological effects of Annexin A1. A synthetic table or figure of the effects/therapeutic potential of AnnexinA1 focused on mechanisms rather than pathology would help.

The Figures were intended to summarize major findings for each condition in order to better orient the readership to the content emphasis of the manuscript; otherwise, it would be cumbersome to follow the manuscript.  However, in light of your comments and those of other reviewers, I now provide 3 Tables to highlight the reported therapeutic potential of annexin-A1 modulation along with relevant citations.  I hope this addresses the problems with original figures.   In addition, I have included Figure 2 summarizing reported mechanisms of action of annexin-A1 modulation in the kidney and the cardiovascular system. 

4- The Line 183-184 “Thus, development of a panel composed of multiple antibodies could serve as novel surrogate biomarker of LN”. The author should clarify how the development of a panel of antibodies would be a biomarker or rephrase this sentence to make it clearer.

This aspect is addressed in the relevant section. 

5- Throughout the text, they are several convoluted sentences that are difficult to understand as they stand (e.g. Line 198 “It was proposed that these proteins form a platform at the cell membrane of neutrophils, of patients with GPA but not healthy subjects, undergoing apoptosis and account for persistent inflammation and autoimmunity thereby offering the opportunity for novel biomarker discovery and therapeutic option(s) for management of kidney injury in GPA.”). I would suggest a re-reading of the article to make the sentences that require it more intelligible.

I have tried to clarify aspects point out by the reviewer.

 Minor points:

  • DEXA is defined twice. DEXA is now defined one.
  • Throughout the text, the naming of Annexin 1/I should be homogenized.

The intent has been to faithfully report investigators’ work in this review article (i.e., nomenclature used by earlier studies).  Nonetheless, to the extent feasible, this aspect has been addressed.

  • There are several typos (e.g. Line 96 “exits”, lines 155 ad 552 “anenxin”, line 205 “collage IV”…) that I did not list exhaustively. A careful reading of the review will allow for correcting these spelling errors.

I do regret the typos but do thank the reviewer for pointing them out.  The submission has been further edited to correct errors/typos. 

  • Line 84 : “Others reported four annexins (with apparent molecular masses of about 32, 35 and 67 kDa” : only 3 apparent molecular masses are listed, is this a mistake ?

This aspect is now addressed (i.e., MW of 32, 35, 45 and 67 kDa) in the relevant section. 

Reviewer 3 Report

In this review, Mozaffari M.S. described the effect of the endogenous glucocorticoid-induced protein Annexin A1 and its therapeutic potential to treat renal and cardiovascular disorders. The manuscript is balanced and well written. However, the figures need substantial improvements. I have suggestions for adjustments, listed below.

  1. Both figures have poor resolution. The figures are hard to follow and the reader can miss important information.
  2. In the figure 1, I suggest to describe the findings in a scheme or in a table (including the respectively citation for each finding). Also, the physiological functions and localization of Annexin A1 in the kidney should be addressed separately from the pharmacological effects in pathological conditions (maybe the figure can be divided in 2 panels).
  3. The figure 2 can be improved. I suggest to include the respectively citation for each finding and use a more detailed scheme or a table.

Author Response

I am grateful to the reviewers for their kind remarks regarding this submission and for their very helpful/insightful comments and suggestion.  I have carefully considered all comments in revision of the manuscript.  The changes in response to reviewers’ comments are highlight in BLUE.

Comments and Suggestions for Authors

In this review, Mozaffari M.S. described the effect of the endogenous glucocorticoid-induced protein Annexin A1 and its therapeutic potential to treat renal and cardiovascular disorders. The manuscript is balanced and well written. However, the figures need substantial improvements. I have suggestions for adjustments, listed below.

Both figures have poor resolution. The figures are hard to follow and the reader can miss important information.

 Given your comments and those of other reviewers, the revised manuscript incorporates Tables (with relevant citations) and Figures that should be of improved quality. 

In the figure 1, I suggest to describe the findings in a scheme or in a table (including the respectively citation for each finding). Also, the physiological functions and localization of Annexin A1 in the kidney should be addressed separately from the pharmacological effects in pathological conditions (maybe the figure can be divided in 2 panels).

I have addressed your valuable comments in the revised submission (i.e., Figures 1-2 and Tables 1-4).

The figure 2 can be improved. I suggest to include the respectively citation for each finding and use a more detailed scheme or a table.

I have addressed your valuable comment by inclusion of Figure 2 summarizing relevant studies and citations are included. 

Round 2

Reviewer 1 Report

The author followed the suggestions and modified the manuscript accordingly. The new presentation of figures and tables made their content more understandable, however resolution (mainly of fig. 2) still needs adjustment.

The paper below should be cited within the topic "III. Recent developments on the role of annexin-A1 in the cardiovascular system".

Jianmin Chen, Lucy V. Norling, Jose Garrido Mesa, Marina De Paula Silva, Sophie E. Burton, Chris Reutelingsperger, Mauro Perretti, Dianne Cooper. Annexin A1 attenuates cardiac diastolic dysfunction in mice with inflammatory arthritis.

Reviewer 2 Report

I thank the author for considering my suggestions and addressing the specific points raised. The Figure 2 illustrates more clearly the interplay between Annexin A1 and renal conditions, and the tables are also useful. Tables and figures still are of poor quality, limiting their legibility. I hope the author will be able to solve this issue for readers’ interest.

Author Response

Dear Reviewer;

I am truly grateful for your comment regarding continued poor resolution of Figures and Tables which I agree with after examining the PDF posted on the journal's website for reviewers' assessment. The fact is that I have converted the WORD version to PDF and notice that it is of good quality and clarity (in my humble opinion). The clearer version has been uploaded to the system.

Kind regards, mm

Reviewer 3 Report

No further comments, the manuscript was substantially improved.

Round 3

Reviewer 2 Report

Thank you for upgrading the figure and table legibility.

Best regards